# Sandalwood Sesquiterpene (*Z*)-α-Santalol Exhibits In Vivo Efficacy Against *Madurella mycetomatis* in *Galleria mellonella* Larvae

**DOI:** 10.3390/molecules30204090

**Published:** 2025-10-15

**Authors:** Shereen O. Abd Algaffar, Stephan Seegers, Shaoqin Zhou, Prabodh Satyal, William N. Setzer, Thomas J. Schmidt, Sami A. Khalid

**Affiliations:** 1Faculty of Pharmacy, University of Science and Technology, Omdurman 14411, Sudan; phd_sh086@hotmail.com; 2University of Münster, Institute of Pharmaceutical Biology and Phytochemistry (IPBP), PharmaCampus, Corrensstrasse 48, D-48149 Münster, Germany; s_seeg03@uni-muenster.de; 3Radboudumc—CWZ Centre of Expertise for Mycology, 6525 Nijmegen, The Netherlands; shaoqin.zhou@radboudumc.nl; 4Aromatic Plant Research Center, Lehi, UT 84043, USA; 5Department of Chemistry, University of Alabama in Huntsville, Huntsville, AL 35899, USA; wsetzer@chemistry.uah.edu

**Keywords:** sandalwood essential oil, Royal Hawaiian sandalwood, *Santalum paniculatum*, (*Z*)-α-santalol, (*Z*)-β-santalol, eumycetoma, *Madurella mycetomatis*, *Galleria mellonella*

## Abstract

In continuation of our recent report on the in vitro activity of sandalwood essential oils and their major components, the sesquiterpenes (*Z*)-α- and -β-santalols, against the pathogenic fungus *Madurella mycetomatis*, causative agent of the neglected tropical disease eumycetoma, we have now tested these isolated constituents, as well as native Royal Hawaiian sandalwood essential oil obtained from *Santalum paniculatum*, for in vivo activity in a model system using infected *Galleria mellonella* larvae. Besides confirming the superior activity of (*Z*)-α-santalol over the (*Z*)*-*β-isomer and the crude essential oil in two further strains of *M. mycetomatis* in vitro, the former compound also turned out to extend the lifespan of the infected larvae significantly, in contrast to (*Z*)-β-santalol, the total oil, or the antifungal drug itraconazole. The present findings not only characterize (*Z*)-α-santalol as a natural compound with promising in vivo activity against eumycetoma but also inspire further studies as a potential for novel, more effective therapy and warranting further studies to understand its mechanism of action and potential clinical applications.

## 1. Introduction

Mycetoma is a Neglected Tropical Disease (NTD) characterized by large subcutaneous lesions that primarily affect the lower limbs in more than 80% of cases. It can be caused by either aerobic or anaerobic bacteria (actinomycetoma) or filamentous fungi (eumycetoma) [1]. More than 90 different mycetoma-causative agents have been identified worldwide, with the disease being endemic in tropical and subtropical countries between latitudes 15° S and 30° N. This region is commonly referred to as the “mycetoma belt”, a term that signifies the high prevalence of the disease in this area [1]. Other countries that have reported cases of mycetoma are the USA, Germany, the Netherlands, Turkey, Lebanon, Saudi Arabia, Iran, the Philippines, Japan, Sri Lanka, and Thailand [2].

Currently, the standard treatment of eumycetoma is a combination therapy with either azole or terbinafine and surgery. In 2022, a WHO survey revealed that 85% of the respondents use itraconazole to treat eumycetoma [3], followed by terbinafine (48%), voriconazole (41%), and posaconazole (33%). Voriconazole and posaconazole are used mainly in high-income countries [4]. Itraconazole, the current first-line antifungal treatment in low-income and middle-income countries, is thought to be only about 40% effective [5]. Furthermore, itraconazole therapy often results in suboptimal outcomes, including prolonged treatment duration, adverse effects, and frequent recurrence [6].

Interestingly, various sandalwood essential oils (EOs) have exhibited antifungal activity against various pathogenic fungi, including *Madurella mycetomatis*, *Aspergillus niger*, *Candida albicans*, and *Cryptococcus neoformans* [7,8,9]. α-Santalol and β-santalol, the major sesquiterpene alcohols in the EOs of *Santalum album* and *S. paniculatum,* have been shown to exhibit antifungal activity against *Madurella mycetomatis* [7] as well as dermatophytic fungus *Trichophyton rubrum*, accompanied by an antimitotic effect [8].

In the present study, we have extended our recent in vitro tests of the isolated sesquiterpenes, as well as the native Royal Hawaiian sandalwood oil (the essential oil of *Santalum paniculatum*), to two additional strains of *M. mycetomatis*. The in vivo toxicity and efficacy of the isolated compounds and the total EO were tested in the established *Galleria mellonella* larvae model.

The established in vivo model, using the invertebrate wax moth larvae *Galleria mellonella*, has proven to be a suitable platform for testing the toxicity and efficacy of compounds that showed promising in vitro results against eumycetoma-causative agents [10,11]. Azole antifungals (Itraconazole, voriconazole, posaconazole, ketoconazole), amphotericin B, and terbinafine have been successfully evaluated using this model [12,13,14].

Interestingly, some essential oils such as *Geigeria alata*, *Myristica fragrans*, *Pimpinella anisum*, *Syzygium aromaticum*, and *Thymus vulgaris* exhibited a wide-spectrum dual in vitro antimycetomal activity against *M. mycetomatis* and *A. madurae* strains (MICs ranging from 0.004 to 0.25% *v*/*v*). Essential oils *M. fragrans* and *P. anisum* prolonged larval survival in infected models with no toxicity [15]. Additionally, CIN-102 (a synergistic mixture based on Cinnamon essential oil) demonstrated activity against fungal biofilms in *A. fumigatus*, *Fusarium solani*, *F. dimerum*, and *Scedosporium apiospermum* at sub-MICs [16] and, in particular, the eumycetoma grain [10].

*G. mellonella* larvae possess a well-developed innate immune system that shares several structural and functional similarities with the mammalian innate immune response. This includes hemocytes that function analogously to mammalian phagocytes, as well as the production of antimicrobial peptides, reactive oxygen species, and melanin, all of which contribute to their immune defense [17,18]. The temperature tolerance of larvae (up to 37 °C) makes them suitable for studying human pathogens under physiologically relevant conditions [19]. The model has been widely validated for assessing virulence and antifungal drug efficacy against a broad range of clinically relevant fungal pathogens, including *Candida albicans*, *Cryptococcus neoformans*, *Aspergillus fumigatus*, and *Fusarium* species [20,21]. The survival of larvae post-infection, often monitored via Kaplan–Meier curves, is a reliable readout of fungal virulence and treatment success. The larvae of *Galleria mellonella* (greater wax moth) have become a well-established invertebrate host model for investigating fungal pathogenesis and for preclinical screening of antifungal agents. Their use is especially advantageous for early-phase research into neglected tropical diseases such as eumycetoma caused by *Madurella mycetomatis*, due to their affordability, ethical acceptability, and biological relevance [10,11,15,22].

## 2. Results and Discussion

### 2.1. In Vitro Activity Against Different Strains of Madurella mycetomatis

Following our previous findings of the antimycetomal activity of sandalwood EOs and isolated santalols on *Madurella mycetomatis* [7], we tested the most active EO from our previous report, “Royal Hawaiian” sandalwood EO (originating from *Santalum paniculatum;* sample denoted Ess-EO5), along with its two isolated major constituents, (*Z*)-α-santalol and (*Z*)-β-santalol (Figure 1), for in vitro activity against two further clinical isolates of the target fungus *M. mycetomatis*, namely, MM58 and P2. The Minimum Inhibitory Concentrations (MICs) are reported in Table 1. It turned out that the two mentioned strains were somewhat less susceptible to the total EO than strain SAK-E07, which was tested earlier [7]. Still, the activity of the two isolated santalols was the same against all three fungal strains tested so far. Thus, as reported previously, (*Z*)-α-santalol was confirmed to be the more active of the two isolated isomers.

### 2.2. In Vivo Activity Against Madurella mycetomatis Infection in Galleria mellonella Larvae

To obtain insight into possible in vivo efficacy and applicability against the fungal mycetoma *M. mycetomatis*, the Royal Hawaiian sandalwood EO Ess-EO5 (*S. paniculatum*) and its two isolated main sesquiterpenes, the (*Z*)-santalols, as well as the standard antifungal drug, itraconazole, were tested on *Galleria mellonella* larvae infected with *M. mycetomatis* strain MM58. Each of the tested substances was administered to the infected larvae by injection via the prolegs on three consecutive days. In each case, three different concentrations were tested by injecting 20 µL/larva of solutions containing 64, 16, and 4 µg/mL of Ess-EO5 or the (*Z*)-santalols at 110, 55, and 13.75 µg/mL, respectively. The larval survival was monitored for 10 days. The drug doses were selected based on the in vitro MIC values of the different samples. Doses higher than, lower than, and in addition to the MIC are usually tested. A control group was treated with itraconazole (5.71 µg/kg). The results of uninfected as well as infected larvae treated only with solvent (PBS) as controls are reported as Kaplan–Meier survival curves in Figure 2.

It is evident from the survival curves of efficacy that neither the antifungal drug itraconazole nor native Royal Hawaiian sandalwood EO caused a significant enhancement of survival in the infected larvae compared to the untreated control (Figure 2A). However, isolated (*Z*)-β-santalol at the highest tested concentration (110 µg/mL) led to a significant decrease in surviving larvae (** *p* = 0.005; Figure 2C) in comparison with the untreated infected controls. In contrast, the same dose of the isomer (*Z*)-α-santalol significantly prolonged the survival of treated larvae (* *p* = 0.014; Figure 2B) compared to that observed in the untreated group.

Kim et al. (2017) reported that α-santalol and β-santalol are fungistatic, similar to grisefulvin, possibly by mechanisms that interfere with the synthesis of certain components of the fungal cell wall, such as chitin [8]. Underlying antifungal activity against *M. mycetomatis* might be attributed to a similar mechanism.

The survival data obtained by treatment of uninfected larvae with the same concentrations of the test substances (Figure 2D–F) show that (*Z*)-β-santalol at its highest tested concentration causes some toxic signs in the larvae (Figure 2F) that counteract the compound’s antifungal activity. (*Z*)-α-Santalol, on the other hand, exhibits a lesser degree of toxicity to the larvae (Figure 2E), which appears to be outmatched, at the same time, by its prompt antifungal effect consistent with the in vitro data (Table 1).

Previous studies on the toxicological effects of sandalwood EO and α-santalol in experimental animals have revealed that the acute oral toxicity (LD_50_) of sandalwood EO in rats is reported to be 5.58 g/kg of body weight. The acute dermal toxicity (LD_50_) of sandalwood oil in rabbits was reported as >5 g/kg of body weight. The acute oral LD_50_ of the principal constituent of sandalwood EO, α-santalol, in rats was reported as 3.8 g/kg. The acute dermal LD_50_ of α-santalol in rabbits was reported as >5 g/kg of body weight [23].

As far as the label of santalol being “safe to use” is concerned, toxicity evaluation of santalol by machine learning methods using a pkCSM server was performed by previous authors. All the predicted outcomes obtained fell under the category of safe to use, which correlates with the origin of the natural source, i.e., sandalwood EO [24].

It is interesting to note that the two santalol isomers, despite their relatively subtle differences in chemical structure (Figure 1), exhibit notable differences in activity against *M. mycetomatis* strains in vitro and their efficacy in the *G. mellonella* in vivo model.

On the backdrop of the relative amounts of α- and β-santalols in the investigated EO, which were previously determined to represent about 46 and 22% of the total EO [7], i.e., to occur in a ratio of about 2:1, it appears that the relatively moderate in vivo effect observed with the oil is due to counteracting antifungal activity and toxic effects of the major and minor isomers, respectively. This underscores the potential for further studies on the antimycetomal effects of purified (*Z*)-α-santalol.

## 3. Materials and Methods

### 3.1. Tested Materials

The Royal Hawaiian sandalwood essential oil “Ess-EO5” tested in this study was identical to the sample analyzed and tested in our previous communication [7]. The isolation of (*Z*)-α-santalol and (*Z*)-β-santalol from this EO was also described in [7]. Their identity and purity (97 and 82%, respectively) were proven by GC-MS analyses [7], NMR data (see Appendix A; all data in accordance with the literature [25,26]) as well as polarimetric analyses ([a]^20^_D_: +11.7° (*c* = 2.5, MeOH) and −51.9° (*c* = 3.0, MeOH) for (+)-(*Z*)-a-santalol and (−)-(*Z*)-b-santalol, respectively; in accordance with the literature [26,27]). Itraconazole was obtained commercially from (I6657, Sigma-Aldrich, St. Louis, MO, USA).

### 3.2. Biological Tests

#### 3.2.1. Cultivation of *Madurella mycetomatis* Strains

*M. mycetomatis* strains MM58 and P2, used in this study, were previously obtained from mycetoma patients and identified to species level [28,29]. They were kept and maintained at the Fungal Culture Collection of the Mycology Reference Laboratory at Radboud University Medical Center (Radboudumc), Nijmegen, The Netherlands. *M. mycetomatis* strains were cultured on Sabouraud Dextrose Agar (SDA) plates and were propagated at 37 °C for 2–3 weeks.

#### 3.2.2. In Vitro Activity Against *M. mycetomatis*

The hyphal suspensions of *M. mycetomatis* strains (MM58 and P2) in RPMI 1640 (Biowest, Nuaillé, France) were sonicated for 10 s (Vevor, High Speed Homogenizer FSH-2A, Changzhou Yineng Experimental Instrument Factory, Changzhou, China), centrifuged at 2600× *g* for 5 min (Andreas Hettich GmbH, Tuttlingen, Germany), and incubated for 7 days at 37 °C. After one week, the mycelia were washed and resuspended in fresh RPMI 1640 medium to give a fungal suspension of 68% to 72% transmission at 660 nm (GENESYS 30 Visible Spectrophotometer, Thermo Fisher Scientific, Waltham, MA, USA). The minimum inhibitory concentrations (MICs) were determined using the microdilution method [30]. A 1:2 serial dilution of test compounds dissolved in dimethyl sulfoxide (DMSO; 472301, Sigma-Aldrich, St. Louis, MO, USA) was prepared in a 96-well microtiter plate. The *M. mycetomatis* strains were subjected to Royal Hawaiian Sandalwood (RHS) essential oil in a series of two-fold dilutions at concentrations from 128 to 4 µg/mL, and to α- and β-santalols in two-fold dilutions from 220 to 6.88 µg/mL. The standard antifungal agent, itraconazole, was used as a positive control and tested in a two-fold dilution series from 1 to 0.03 µg/mL. To each well, 100 µL of adjusted fungal suspension and 1 µL of the test compound were added, followed by 1 µL of resazurin to give a final concentration of 0.15 mg/mL. The plates were sealed and incubated at 37 °C for 7 days. The assay plates were inspected on day 7 for visual and spectrometric endpoints. For spectrophotometric MICs, absorbance was measured at 600 nm (Anthos HT3 Reader, Wals, Austria). All assays were performed in three independent replicates [30].

#### 3.2.3. In Vivo Studies in the *Galleria mellonella* Model

*Galleria mellonella* larvae were purchased from the Blue Lagoon Pets Company, Maassluis, The Netherlands. Larvae were kept in the dark at 21 °C, a temperature corresponding to room temperature. Active larvae, without any discoloration and weighing 300–500 mg were used for the experiments.

#### 3.2.4. In Vivo Efficacy of Royal Hawaiian Sandalwood (*Santalum paniculatum*) Essential Oil and Pure Santalols in the *Galleria mellonella* Model

In short, mycelia from *M. mycetomatis* strain (MM58) were obtained from 2-week-old cultures grown on Sabouraud agar plates, scraped from the plate, and inoculated in RPMI 1640 culture medium, 20 mM morpholinepropanesulfonic acid (MOPS; M1254-2506, Sigma-Aldrich, New Taipei City, Taiwan), and chloramphenicol (100 mg/L; Oxoid, Basingstroke, UK). After 2 weeks of incubation at 37 °C, the hyphae were collected by vacuum filtration through a 0.22 µm filter (Corning 430513; Corning Incorporated, Corning, NY, USA) and washed with phosphate-buffered saline (PBS). The wet biomass was scraped from the filter, weighed, resuspended in PBS, and sonicated for 2 min at 28 microns, followed by centrifugation at 3400 rpm for 5 min. The biomass was adjusted with PBS to a concentration of 1 g/10 mL. A 40 µL of this inoculum was injected into the left last proleg of the larvae with an insulin 29 G U-100 needle (BD Diagnostics, Franklin Lakes, NJ, USA), resulting in a final inoculum of 4 mg/larvae. To exclude any contamination during the preparation of the inoculum, 10 μL of each inoculum prepared was inoculated on Sabouraud and blood agar plates. A total of 15 larvae per group were used, and three biological replicates were performed. Each infected larva was treated with 20 µL doses of either Royal Hawaiian Sandalwood EO at final concentrations of 4, 16, 64 µg/mL or α- and β-santalols at final concentrations of 13.75, 55, and 110 µg/mL for three consecutive days at 4, 28, and 52 h post-infection. Additionally, 5.71 mg/kg itraconazole and solvent-only control groups (5% DMSO in PBS) were included. Larvae were then incubated at 37 °C, and their survival was monitored and recorded for 10 days [10,15].

#### 3.2.5. In Vivo Toxicity Studies of Royal Hawaiian Sandalwood (*Santalum paniculatum*) Essential Oil and Pure Santalols in the *Galleria mellonella* Model

To determine the toxicity profiles of Royal Hawaiian Sandalwood EO and α- and β- santalols, 45 *G. mellonella* larvae were treated simultaneously as the infected groups. Each larva was injected at day 0 via the left last proleg with a 20 µL dose containing final concentrations of 4, 16, 64 µg/mL of EO and 13.75, 55, and 110 µg/mL of α- and β-santalols and monitored for 10 days. Larvae injected with solvent-only (5% DMSO in PBS) were used as negative controls. Experiments were performed in triplicate [10,15].

#### 3.2.6. Statistical Analysis

Kaplan–Meier survival curves were generated to evaluate the survival rates of uninfected and infected *Galleria mellonella* larvae. The Log-rank test was performed using GraphPad Prism 10 (version 10.4.1, GraphPad Inc., Boston, MA, USA) to compare the survival curves and determine if there was a statistically significant difference between the treatment groups. A *p*-value less than 0.05 was considered statistically significant [15].

## 4. Conclusions

This study builds upon our previous in vitro findings by demonstrating that (*Z*)-α-santalol, a primary constituent of sandalwood EOs, including Royal Hawaiian sandalwood EO from *Santalum paniculatum*, exhibits consistent antifungal activity against multiple *Madurella mycetomatis* strains in vitro. We could now demonstrate that this sesquiterpene also possesses significant in vivo efficacy in a *Galleria mellonella* infection model. Among the compounds tested—including both (*Z*)-α-santalol and (*Z*)-β-santalol, the crude Royal Hawaiian sandalwood essential oil (Ess-EO5), and the standard antifungal itraconazole—only (*Z*)-α-santalol significantly prolonged the survival of the infected larvae. In contrast, (*Z*)-β-santalol showed limited efficacy and demonstrated larval toxicity at higher concentrations.

The favorable therapeutic profile of (*Z*)-α-santalol is underscored by its unique combination of potent antifungal activity and minimal in vivo toxicity. The lack of therapeutic benefit observed with the crude essential oil is likely due to the opposing effects of its constituent santalol isomers, highlighting the necessity of isolating and utilizing the bioactive α-isomer in future formulations.

These findings position (*Z*)-α-santalol as a promising natural compound for further development as an antifungal agent targeting *M. mycetomatis* and potentially eumycetoma more broadly. However, additional studies are crucial to fully understand its precise mechanism of action, assess its pharmacokinetic and pharmacodynamic profiles, and evaluate its efficacy in mammalian models. Given the urgent need for more effective and accessible treatments for this neglected tropical disease, (*Z*)-α-santalol represents a promising candidate for advancing the development of nature-derived antimycetomal drugs.

## Figures and Tables

**Figure 1 molecules-30-04090-f001:**
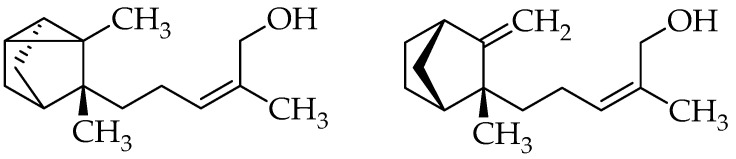
Structures of (*Z*)-α-santalol (**left**) and (*Z*)-β-santalol (**right**).

**Figure 2 molecules-30-04090-f002:**
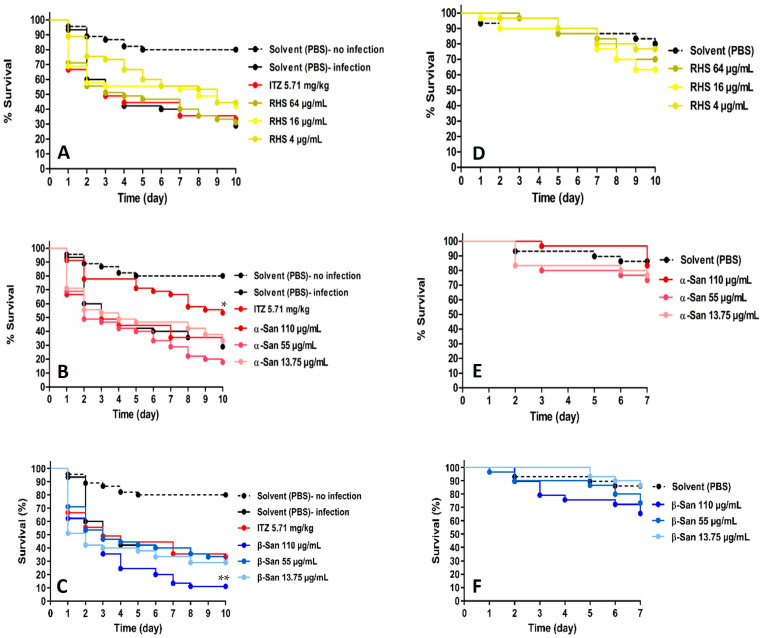
Kaplan–Meier survival curves of *Galleria mellonella* larvae over 7–10 days. (**A**–**C**) Dotted lines indicate uninfected larvae which served as solvent controls and which were treated with PBS. The solid lines indicate infected larvae treated with 20 µL of the test substances at the mentioned concentrations at 4, 28, and 52 h post-infection. (**D**–**F**) Toxicity to uninfected larvae. Dotted lines: uninfected larvae as solvent controls, treated with PBS; Solid lines: uninfected larvae treated with 20 µL single dose of the test substances in the mentioned concentrations. (**A**,**D**) RHS = Royal Hawaiian sandalwood EO; (**B**,**E**) (*Z*)-α-santalol; (**C**,**F**) (*Z*)-β-santalol; ITZ: itraconazole. * *p* < 0.05; ** *p* < 0.01.

**Table 1 molecules-30-04090-t001:** MIC values of Royal Hawaiian sandalwood essential oil Ess-EO5 (*S. paniculatum*), its isolated main constituents, (*Z*)-α- and β-santalols, as well as the fungistatic itraconazole as positive control against various *M. mycetomatis* strains. Data for strain SAK-E07 are from our previous report [7] and included here for comparison.

Sample	MIC (µg/mL)		
	SAK-E07	MM58	P2
Ess-EO5	16	64	64
(*Z*)-α-santalol	27.5	27.5	27.5
(*Z*)-β-santalol	55	55	55
Itraconazole	0.25	0.25	0.062

## Data Availability

The original contributions presented in this study are included in the article/Appendix A. Further inquiries can be directed to the corresponding authors.

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
