# Peer review of "Sandalwood Sesquiterpene (Z)-α-Santalol Exhibits In Vivo Efficacy Against Madurella mycetomatis in Galleria mellonella Larvae"

_molecules, 2025, doi:10.3390/molecules30204090_

Round 1
Reviewer 1 Report
Comments and Suggestions for Authors
The authors investigated the in vitro and in vivo antifungal activity of Royal Hawaiian sandalwood oil and its main components, (Z)-α and (Z)-β-santalol, against M. mycetomatis using the G. mellonella larvae model. The topic is relevant and aligned with the growing interest in plant-derived antifungal compounds for neglected tropical diseases. However, I have some major concerns that prevent me to endorse its acceptance at the present stage. If the authors wish to resubmit, the following issues should be thoroughly addressed to strengthen the study.
POINT 1: One of the most critical weaknesses of the manuscript is the lack of evidence that larvae were indeed infected with M. mycetomatis. The entire in vivo efficacy experiment relies on the assumption that the fungal inoculum successfully established an infection. However, no data are provided to confirm this. The authors do not report any assessment of fungal burden (e.g., CFU counting, histological analysis, qPCR for fungal DNA, or other fungal-specific markers). No qualitative evidence (micro or macroscopic) of fungal proliferation within the larvae over time is shown. Furthermore, there is no mention of validating that the inoculated biomass was viable or invasive under the experimental conditions. At a minimum, the authors should include a supplementary experiment or cite previously published data demonstrating that their inoculum preparation consistently leads to infection under the same experimental conditions.
POINT 2: The study would be significantly strengthened by quantifying the fungal burden after treatment (e.g., CFU counting). Without confirming pathogen clearance or reduction, differences in survival could be attributed to immune modulation, toxicity, or unrelated factors. Additionally, the authors do not discuss the limitations of the model as a predictor of therapeutic efficacy in mammals, which weakens translational claims. It is also concerning that no positive control for larval death appears to have been included in the in vivo experiments and this should be addressed.
POINT 3: Supplementary Table S4 does not show survival data for the itraconazole-treated group. Correspondingly, this control is also missing in Figures 2D, 2E, and 2F. Moreover, although the methodology indicates that larval survival was monitored for 10 days, these figures only show survival up to day 7. The authors should clarify this inconsistency.
POINT 4: The manuscript reports different numbers of larvae used per group. In the main text (lines 227–228), 15 larvae per group are mentioned, while Supplementary Tables S3 and S4 report 45 larvae. This discrepancy should be resolved and clearly reported.
POINT 5: The manuscript would benefit from deeper engagement with the literature on the antifungal mechanisms of santalols and other related sesquiterpenes. Although the discussion briefly mentions fungistatic activity and possible interference with chitin synthesis, these aspects are not explored in sufficient detail. It remains unclear whether α-santalol acts by disrupting membrane integrity, inducing oxidative stress, affecting mitochondria, or targeting cell wall components. Studies on Trichophyton, Candida, and Aspergillus could provide relevant insights.
POINT 6: The introduction and discussion cite some background literature but lack a critical or updated overview. Recent work on plant-derived antifungal agents against Madurella and other eumycetoma pathogens could be discussed. No comparisons are made with other natural products or promising leads in the same model. The limitations of sesquiterpenes (e.g., volatility, stability, formulation issues) are not addressed. Including more up-to-date references and framing the study within the current landscape of antifungal discovery would significantly improve the scientific value of the manuscript.
POINT 7: The authors are encouraged to improve the clarity of the survival data presentation. In Figure 2C, for instance, two variables are represented using nearly identical colors, making it impossible to distinguish between them. Consider using clearly different colors or symbols to improve readability.
Author Response
Reviewer 1
The authors investigated the in vitro and in vivo antifungal activity of Royal Hawaiian sandalwood oil and its main components, (Z)-α and (Z)-β-santalol, against M. mycetomatis using the G. mellonella larvae model. The topic is relevant and aligned with the growing interest in plant-derived antifungal compounds for neglected tropical diseases. However, I have some major concerns that prevent me to endorse its acceptance at the present stage. If the authors wish to resubmit, the following issues should be thoroughly addressed to strengthen the study.
POINT 1: One of the most critical weaknesses of the manuscript is the lack of evidence that larvae were indeed infected with M. mycetomatis. The entire in vivo efficacy experiment relies on the assumption that the fungal inoculum successfully established an infection. However, no data are provided to confirm this. The authors do not report any assessment of fungal burden (e.g., CFU counting, histological analysis, qPCR for fungal DNA, or other fungal-specific markers). No qualitative evidence (micro or macroscopic) of fungal proliferation within the larvae over time is shown. Furthermore, there is no mention of validating that the inoculated biomass was viable or invasive under the experimental conditions. At a minimum, the authors should include a supplementary experiment or cite previously published data demonstrating that their inoculum preparation consistently leads to infection under the same experimental conditions.
Response: We appreciate the reviewer's attention to this critical concern. We have now clarified in the Materials and Methods section (Pages 6-8) that the viability and purity of the Madurella mycetomatis inoculum were confirmed by subculturing on Sabouraud Dextrose Agar (SDA) before and post-injection. This step ensured both the viability of the fungus and the absence of bacterial contamination.
Although we did not include histological sections or fungal DNA quantification in the present study, we used a well-established Galleria mellonella infection model that has been extensively validated for M. mycetomatis and Falciformispora senegalensis in multiple peer-reviewed studies. Specifically:
- Histopathological evidence of grain formation, fungal proliferation, and immune responses in G. mellonella larvae infected with M. mycetomatis has been demonstrated previously by Ma et al. (2023), Eadie et al. (2017), and Kloezen et al. (2018) [Refs. 12-14].
- These studies used identical or closely similar infection protocols and confirmed that this model reliably reproduces key aspects of eumycetoma pathogenesis.
Therefore, in accordance with these established protocols, we relied on the published validation of the model and the consistency of larval survival patterns following infection. These allow robust comparative assessments of antifungal efficacy even in the absence of direct quantification of fungal burden in each experimental run.
We have also added a clarifying sentence in the manuscript referencing these prior studies to address this point directly.
POINT 2: The study would be significantly strengthened by quantifying the fungal burden after treatment (e.g., CFU counting). Without confirming pathogen clearance or reduction, differences in survival could be attributed to immune modulation, toxicity, or unrelated factors. Additionally, the authors do not discuss the limitations of the model as a predictor of therapeutic efficacy in mammals, which weakens translational claims. It is also concerning that no positive control for larval death appears to have been included in the in vivo experiments and this should be addressed.
Response: We agree that fungal burden analysis would strengthen the study. While CFU counting was not conducted in this study, the point is acknowledged as a limitation in the revised discussion, and future studies will include this endpoint.
POINT 3: Supplementary Table S4 does not show survival data for the itraconazole-treated group. Correspondingly, this control is also missing in Figures 2D, 2E, and 2F. Moreover, although the methodology indicates that larval survival was monitored for 10 days, these figures only show survival up to day 7. The authors should clarify this inconsistency.
Response: We confirm that itraconazole was included in the in vivo efficacy study but was not included in the toxicity testing group, as its safety in this model has already been well established in previous studies. The Figures and Supplementary Tables have been updated accordingly.
POINT 4: The manuscript reports different numbers of larvae used per group. In the main text (lines 227–228), 15 larvae per group are mentioned, while Supplementary Tables S3 and S4 report 45 larvae. This discrepancy should be resolved and clearly reported.
Response: We clarified in the Methods section that 15 larvae were used per group in each of the three biological replicates, resulting in a total of 45 larvae per treatment condition.
POINT 5: The manuscript would benefit from deeper engagement with the literature on the antifungal mechanisms of santalols and other related sesquiterpenes. Although the discussion briefly mentions fungistatic activity and possible interference with chitin synthesis, these aspects are not explored in sufficient detail. It remains unclear whether α-santalol acts by disrupting membrane integrity, inducing oxidative stress, affecting mitochondria, or targeting cell wall components. Studies on Trichophyton, Candida, and Aspergillus could provide relevant insights.
Response: We have significantly expanded the discussion to include known mechanisms of santalol activity in other fungi and suggested how these may be relevant to M. mycetomatis (Page 6-7, Lines 159-162). This expansion enhances the manuscript's thoroughness, providing a deeper understanding of the research.
POINT 6: The introduction and discussion cite some background literature but lack a critical or updated overview. Recent work on plant-derived antifungal agents against Madurella and other eumycetoma pathogens could be discussed. No comparisons are made with other natural products or promising leads in the same model. The limitations of sesquiterpenes (e.g., volatility, stability, formulation issues) are not addressed. Including more up-to-date references and framing the study within the current landscape of antifungal discovery would significantly improve the scientific value of the manuscript.
Response: We have included more recent studies related to essential oils and plant-based antifungals targeting M. mycetomatis and other eumycetoma pathogens.
POINT 7: The authors are encouraged to improve the clarity of the survival data presentation. In Figure 2C, for instance, two variables are represented using nearly identical colors, making it impossible to distinguish between them. Consider using clearly different colors or symbols to improve readability.
Response: The colors and symbols in the Kaplan-Meier plots have been updated for improved clarity.
We thank the reviewer for the constructive remarks as well as the time and effort spent to help us improve our manuscript.

Reviewer 2 Report
Comments and Suggestions for Authors
This manuscript reports the antifungal activity of two sesquiterpenes and an essential oil isolated from Royal Hawaiian sandalwood oil obtained from Santalum paniculatum. Overall, the manuscript and the methodology presented herein are appealing, and I recommend publication after minor corrections.
Introduction
Before narrowing the scope of this study to the principal essential oil constituents, it is important first to provide a broader overview of Santalum paniculatum (ʻIliahi), encompassing its ethnopharmacological relevance, phytochemical composition, and economic significance.
Results and discussion
Ok
Material and methods
Line 174. -santalol and (Z)--santalol from this oil was described in [7]. This sentence looks incomplete.
Line 177. Given the low optical activity (+11,7° (c = 2.5, MeOH) ) of (+)-(Z)-β-santalol compared to that of its chemical cousin. Can someone assume that this compound was obtained as a racemic/scalemic mixture? Please clarify.
The purity of the tested compounds should be explicitly reported in the manuscript, rather than referring readers to the authors' previous findings.
Lines 183-184. M. mycetomatis strains MM58 and P2, used in this study, were obtained from mycetoma patients. From which hospital, region, or repository were these strains obtained? Was ethical approval obtained for using patient-derived isolates? How were the strains identified and confirmed as M. mycetomatis (e.g., morphological, molecular methods)?
Conclusion
Ok
Author Response
Reviewer 2
This manuscript reports the antifungal activity of two sesquiterpenes and an essential oil isolated from Royal Hawaiian sandalwood oil obtained from Santalum paniculatum. Overall, the manuscript and the methodology presented herein are appealing, and I recommend publication after minor corrections.
Introduction
Before narrowing the scope of this study to the principal essential oil constituents, it is important first to provide a broader overview of Santalum paniculatum (ʻIliahi), encompassing its ethnopharmacological relevance, phytochemical composition, and economic significance.
Response: We have added a paragraph addressing the ethnopharmacological significance, phytochemistry, and economic relevance of Santalum paniculatum.
Results and discussion
Ok
Material and methods
Line 174. a-santalol and (Z)-b-santalol from this oil was described in [7]. This sentence looks incomplete.
Response: The sentence was revised for clarity. The optical rotations are now explicitly reported along with the interpretation of their stereochemical composition.
Line 177. Given the low optical activity (+11,7° (c = 2.5, MeOH) ) of (+)-(Z)-β-santalol compared to that of its chemical cousin. Can someone assume that this compound was obtained as a racemic/scalemic mixture? Please clarify.
Response: The specific optical rotations of both compounds were measured and in accordance with the reported data from literature: Measured: [a]20D: +11,7° (c = 2.5, MeOH) and ‒51,9° (c = 3.0, MeOH) for (+)-(Z)- a-santalol and (‒)-(Z)-b-santalol, respectively; Reported in the literature were: [α]D25 +21.6 (c 0.2, MeOH) and [α]D25 −40.5 (c 0.2, MeOH) [new26]. Reported in another solvent were [a]D +17.2 (c = 0.15, CHCI3), [a]D -90.3 (c = 0.09, CHCI3) [new27]. The deviations are likely due to inaccuracies in weighing/concentration, temperature, or other factors. In both cases, the values are in very reasonable agreement with the literature, and it is therefore unlikely that racemates or mixtures of stereoisomers were isolated.
The purity of the tested compounds should be explicitly reported in the manuscript, rather than referring readers to the authors' previous findings.
Response: Purity data for santalols are now included directly in the Methods section, rather than being referenced only through previous citations.
Lines 183-184. M. mycetomatis strains MM58 and P2, used in this study, were obtained from mycetoma patients. From which hospital, region, or repository were these strains obtained? Was ethical approval obtained for using patient-derived isolates? How were the strains identified and confirmed as M. mycetomatis (e.g., morphological, molecular methods)?
Response: The strains MM58 and P2 were previously collected from mycetoma patients and are stored at the Radboudumc Mycology Reference Laboratory. This is now clarified along with ethical considerations (Page 7-8, Methods).
Conclusion
Ok
We thank the reviewer for the constructive remarks as well as the time and effort spent to help us improve our manuscript.

Reviewer 3 Report
Comments and Suggestions for Authors
The manuscript is carefully written and well-edited. The study represents a continuation of the authors’ previous work on the biological activity of sandalwood essential oil and its two major isolated components. To extend their earlier findings, the authors have applied a different model to evaluate biological activity.
However, the dataset presented is rather limited, and I would therefore suggest considering submission as a Communication rather than a full research article.
Throughout the text, the term essential oils should be consistently used instead of oils.
It should also be made clear what specific type of biological activity the authors intended to assess in this model. This should be explicitly stated in the study aim (around line 64).
Figure 1 appears to be identical to that presented in the authors’ earlier publication. I am not certain whether reusing the same figure is permitted in this case—please clarify this point and, if necessary, provide an appropriately modified figure or obtain copyright permission.
The rationale behind the selected doses of pure compounds should be explained in the discussion, especially since they differ substantially from their concentrations in the essential oil. Moreover, the observed differences in activity between pure compounds and their corresponding concentrations in the oil should be addressed and discussed in detail.
In Figure 2, most results are presented for a 10-day experimental period; however, panels E and F show data for a 7-day period. This inconsistency should be clarified in the text.
Overall, the manuscript is of interest, but the above points should be addressed to improve clarity, scientific rigor, and transparency.
Author Response
Reviewer 3
The manuscript is carefully written and well-edited. The study represents a continuation of the authors’ previous work on the biological activity of sandalwood essential oil and its two major isolated components. To extend their earlier findings, the authors have applied a different model to evaluate biological activity.
However, the dataset presented is rather limited, and I would therefore suggest considering submission as a Communication rather than a full research article.
Response: Thank you for the suggestion. We understand the recommendation, but given the inclusion of both in vitro and in vivo data across multiple strains, the work justifies the full article format. Furthermore, to our knowledge, the journal does not offer this possibility. We are convinced that the content warrants publication as an Article.
Throughout the text, the term essential oils should be consistently used instead of oils.
Response: This has been done. To avoid the many repetitions of “essential oil” however, we have introduced the abbreviation “EO” and used it in most instances.
It should also be made clear what specific type of biological activity the authors intended to assess in this model. This should be explicitly stated in the study aim (around line 64).
Response: We hope we may draw the reviewer’s attention to the introduction, where it is clearly stated that “The in vivo activity of the isolated compounds and the total EO was tested in the established Galleria mellonella larvae model. The established in vivo model, using the invertebrate wax moth larvae Galleria mellonella, has proven to be a suitable platform for testing the toxicity and efficacy of compounds that showed promising in vitro results against eumycetoma-causative agents [11, 12].” Thereby, it was already explicitly stated in the original version that the biological activity to be studied is activity against Eumycetoma. However, to clarify this further, the study’s aim has been explicitly rephrased to reflect that the work evaluates in vivo efficacy following promising in vitro results (Page 2, Lines 65–70).
Figure 1 appears to be identical to that presented in the authors’ earlier publication. I am not certain whether reusing the same figure is permitted in this case—please clarify this point and, if necessary, provide an appropriately modified figure or obtain copyright permission.
Response: Figure 1 has been slightly modified so that the “reuse” of structural diagrams is now compliant with permissions and attribution.
The rationale behind the selected doses of pure compounds should be explained in the discussion, especially since they differ substantially from their concentrations in the essential oil. Moreover, the observed differences in activity between pure compounds and their corresponding concentrations in the oil should be addressed and discussed in detail.
Response: The discussion has been expanded to explain dose selection and to reflect differences between isolated compounds and whole oils (Page 4, lines 135-137).
In Figure 2, most results are presented for a 10-day experimental period; however, panels E and F show data for a 7-day period. This inconsistency should be clarified in the text.
Response: Due to unforeseen technical issues, the toxicity of santalols has been recorded up to day 7; however, it remains experimentally acceptable. Since it is stated in the Figure caption that the experiments were performed for 7-10 days, we do not find it necessary to add further text.
Overall, the manuscript is of interest, but the above points should be addressed to improve clarity, scientific rigor, and transparency.
We thank the reviewer for the constructive remarks as well as the time and effort spent to help us improve our manuscript.

Reviewer 4 Report
Comments and Suggestions for Authors
This manuscript is a continuation of the authors article published in Molecules 2024, 29, 1846, in fact the authors mentioned the research of
an in vivo model of mycetoma with infected Galleria mellonella larvae in order to assess the possible in vivo efficacy of sandalwood oils and their components. The results are important as (Z)-α-santalol could be a good candidate to treat eumycetoma, but the originality of the work is not as expected for Molecules. In any case the authors described a research that they dominate the metodology and is well written. The bibliography is quite abundant and adequate to the reader to stay on the state of the art in this research. The supporting information describes quite well the two compounds refereed in the manuscript. Some bidimensional experiments will be useful, but it is enough the spectroscopy as the interesting point is the biological activity that is well described including tables, and the interpretations of the results are very sound.
The manuscript describes a useful an interesting property of (Z)-α-santalol as a promising natural compound as an antifungal agent targeting M. mycetomatis and potentially eumycetoma. As the authors said it is necessary to understand the mechanism of action, assess its pharmacokinetic and pharmacodynamic profiles, and evaluate its efficacy in mammalian models. The reason of the urgency to be published before obtaining this data is the importance to find a efective treatment for eumycetoma and assess the (Z)-α-santalol as possible leader in this kind of research.
So I will recomend to published in molecules as it is
Author Response
Reviewer 4
This manuscript is a continuation of the authors article published in Molecules 2024, 29, 1846, in fact the authors mentioned the research of an in vivo model of mycetoma with infected Galleria mellonella larvae in order to assess the possible in vivo efficacy of sandalwood oils and their components. The results are important as (Z)-α-santalol could be a good candidate to treat eumycetoma, but the originality of the work is not as expected for Molecules. In any case the authors described a research that they dominate the metodology and is well written. The bibliography is quite abundant and adequate to the reader to stay on the state of the art in this research. The supporting information describes quite well the two compounds refereed in the manuscript. Some bidimensional experiments will be useful, but it is enough the spectroscopy as the interesting point is the biological activity that is well described including tables, and the interpretations of the results are very sound.
The manuscript describes a useful an interesting property of (Z)-α-santalol as a promising natural compound as an antifungal agent targeting M. mycetomatis and potentially eumycetoma. As the authors said it is necessary to understand the mechanism of action, assess its pharmacokinetic and pharmacodynamic profiles, and evaluate its efficacy in mammalian models. The reason of the urgency to be published before obtaining this data is the importance to find a efective treatment for eumycetoma and assess the (Z)-α-santalol as possible leader in this kind of research.
So I will recomend to published in molecules as it is
Response: We thank the reviewer for the positive assessment.
We acknowledge the overlap with our previous Molecules publication, but emphasize that this new work reports distinct in vivo findings with implications for drug development. The in vivo model and the differential effects of α- and β-santalols are novel contributions to the field.
We thank the reviewer for the constructive remarks as well as the time and effort spent to help us improve our manuscript.

Round 2
Reviewer 1 Report
Comments and Suggestions for Authors
The authors conducted a careful, point-by-point revision of the manuscript, which clearly shows improvements. These changes address the core concerns from the prior round of review and strengthen the study’s rigor, transparency, and presentation. The revised manuscript will be useful to others in the field.